# Sterile Neutrinos with Neutrino Telescopes

**Carlos A. Argüelles** [1] and **Jordi Salvado** [2,*]

1   Department of Physics & Laboratory for Particle Physics and Cosmology, Harvard University, Cambridge, MA 02138, USA; carguelles@fas.harvard.edu
2   Departament de Física Quàntica i Astrofísica and Institut de Ciencies del Cosmos, Universitat de Barcelona, Diagonal 647, E-08028 Barcelona, Spain
*   Correspondence: jsalvado@fqa.ub.edu

**Abstract:** Searches for light sterile neutrinos are motivated by the unexpected observation of an electron neutrino appearance in short-baseline experiments, such as the Liquid Scintillator Neutrino Detector (LSND) and the Mini Booster Neutrino Experiment (MiniBooNE). In light of these unexpected results, a campaign using natural and anthropogenic sources to find the light (mass-squared-difference around 1 eV$^2$) sterile neutrinos is underway. Among the natural sources, atmospheric neutrinos provide a unique gateway to search for sterile neutrinos due to the broad range of baseline-to-energy ratios, $L/E$, and the presence of significant matter effects. Since the atmospheric neutrino flux rapidly falls with energy, studying its highest energy component requires gigaton-scale neutrino detectors. These detectors—often known as neutrino telescopes since they are designed to observe tiny astrophysical neutrino fluxes—have been used to perform searches for light sterile neutrinos, and researchers have found no significant signal to date. This brief review summarizes the current status of searches for light sterile neutrinos with neutrino telescopes deployed in solid and liquid water.

**Keywords:** neutrino oscillations; neutrino telescopes; sterile neutrinos

## 1. Neutrino Anomalies

In 1996, the LSND experiment published an intriguing result that suggested the possible existence of an extra, relatively light mass state involved in neutrino oscillations [1,2]. The hint of a still undiscovered fermion singlet beyond the standard model strongly motivated follow-up experiments. In 2013, the MiniBooNE experiment, which operated in both neutrino and anti-neutrino modes, also found a result that was inconsistent with the standard three neutrino picture [3] and compatible with LSND. Both LSND and MiniBooNE observed an excess of events with respect to eam background expectation. At the same time, a new calculation of the reactor neutrino fluxes put an overall tension of the total normalization for the reactor neutrino experiments [4]—the result was also consistent with the measurement from the GALLEX Cr-51 source experiment [5]. These overall missing events, both in reactors and radioactive sources, were also understood as evidence for a sterile neutrino.

However, not all results were positive; some of the experiments strongly constrained the sterile parameter space [6–10]. The analysis at the time [11–13] showed a tension between different data sets, but also focused the region of interest for the sterile parameters to $\Delta m_{14}^2 = 1$ eV and $\sin^2 2\theta_{24} = 0.1$. The experimental evidence for sterile neutrinos, and therefore for beyond the standard physics, was also addressed from the theoretical and phenomenological perspective, and different models and phenomenological explanations were proposed [14–30].

From a phenomenological perspective, the main challenge is in accommodating the positive results of the appearance experiments with the constraints from the disappearance experiments. This general statement is unavoidable since the oscillation probability amplitudes in different channels are strongly correlated.

However, constraints from electron- and muon-neutrino disappearances are more susceptible to uncertainties arising from the mismodeling of the neutrino flux. This is not the case of neutrino telescopes, where the sensitivity relies on the fact that the matter potential enhances the disappearance probability at around TeV energies, making these analyses unique. This effect only exists if the observed excess of events is due to oscillation physics and does not apply if, for example, the excess is produced by the decay of a heavy state [14,15,22,24,30,31]. Models proposing this scenario would not be affected by the current IceCube bounds. However, this does not solve all the tension, and today, the global picture is hard to accommodate in a vanilla light sterile neutrino scenario. A more detailed review on the status of light sterile neutrinos can be found in Refs. [32–35].

## 2. Neutrinos Propagating through the Earth with a Sterile State

The presence of matter alters neutrino oscillations; for example, differences between the charged- and neutral-current matter potentials play an important role in solar neutrinos [36,37]. In the standard three-neutrino scenario, the effect of the matter potential is small for Earth-traversing neutrinos at the energies typical of atmospheric and long-baseline neutrino experiments. On the contrary, the Earth's matter effects may significantly enhance the oscillation amplitude between an Earth-traversing standard neutrino and a light, sterile neutrino for neutrinos with energies between 1 and 10 TeV [38–43].

The IceCube Neutrino Observatory, located at the geographic South Pole, is sensitive to neutrinos in this energy range. Most of the neutrinos that IceCube sees at these energies are produced in cosmic-ray interactions with the atmosphere. As a result, these neutrinos are dominantly produced as muon neutrinos. As indicated in Refs. [43–50], IceCube can search for a muon-neutrino disappearance as a signature of a light sterile neutrino.

In Figure 1, we show the disappearance probability for an Earth-traversing muon anti-neutrino. This disappearance probability was calculated with the nuSQuIDS [51], which consistently accounts for oscillation and attenuation. In the $10^3$ GeV–$10^4$ GeV range, the substantial disappearance happens due to the enhancement of the probability amplitude arising from the neutral-current matter potential.

At energies below 100 GeV, a light sterile neutrino gives rise to two effects. The first is a fast oscillation, which results in an overall deficit of events. This deficit is proportional to $\sin^2 2\theta_{24}$; however, this effect is not currently observable due to the large uncertainties on the atmospheric flux normalization. The second is an effective modification of the standard model matter potential [52], which can be seen as a non-standard interaction. Limits on non-standard interactions allow us to put bounds to the $\mathcal{O}(\text{eV})$ sterile neutrinos using atmospheric neutrino data under 100 GeV [7,53].

As will be shown in the following sections, this can also be used to place bounds on heavy sterile using the whole energy spectrum [54–56].

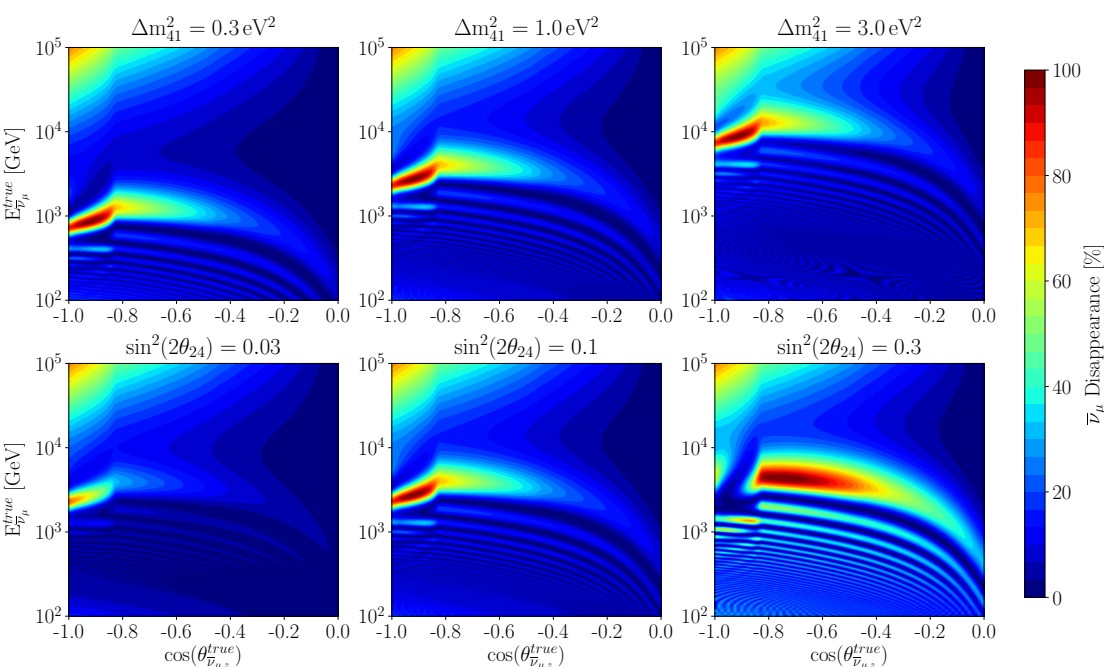

**Figure 1.** The survival probability for muon anti-neutrinos traveling through the Earth. For the top row $\sin^2(2\theta_{24}) = 0.1$ and $\Delta m^2_{24}$ is set to the value shown in the labels, for the bottom row $\Delta m^2_{24} = 1\ \mathrm{eV}^2$ and $\sin^2(2\theta_{24})$ is set to the values in the labels. The calculation of the propagation is done with the nuSQuIDS library [51]. Figure from [55].

## 3. Searches with Atmospheric Neutrinos below 100 GeV in Neutrino Telescopes

The existence of a light sterile neutrino modifies the oscillation probability relevant for atmospheric neutrino oscillations. As such, analyses that aim to measure the standard atmospheric neutrino oscillation parameters are sensitive to distortions induced by a light sterile neutrino. Specifically, the energy and zenith distributions will be modified by a light sterile neutrino. Below 100 GeV, a sterile neutrino with a mass-squared-difference compatible with LSND and MiniBooNE data would produce a subleading modification to the atmospheric parameters.

Thus, an atmospheric analysis would fit the standard parameters along with the light sterile neutrino parameters simultaneously. Furthermore, for a sterile neutrino with a mass above 0.1 eV, neutrino telescopes cannot resolve oscillations, and the mixing elements give rise to the only observable effects. Since the atmospheric neutrino oscillation is predominantly sensitive to conversion between tau neutrinos and muon neutrinos, these experiments are most sensitive to the magnitudes of $U_{\mu 4}$ and $U_{\tau 4}$, with a subleading sensitivity to one of the two new *CP*-violating phases introduced by the standard parameterization [57].

Recent results and projected sensitivities can be seen in Figure 2. Currently, atmospheric neutrino experiments have the strongest limits, though bounds from accelerator-based measurements are competitive in constraining the magnitude of $U_{\tau 4}$. The leading atmospheric neutrino constraints are due to Super-Kamiokande and IceCube-DeepCore (3 years), which are expected to be significantly improved by an upcoming decadal analysis (denoted 'this work'). Interestingly, ANTARES prefers a non-zero $U_{\tau 4}$ solution, though at significantly less than three sigma. This claim is expected to be tested in the aforementioned IceCube-DeepCore decadal analysis.

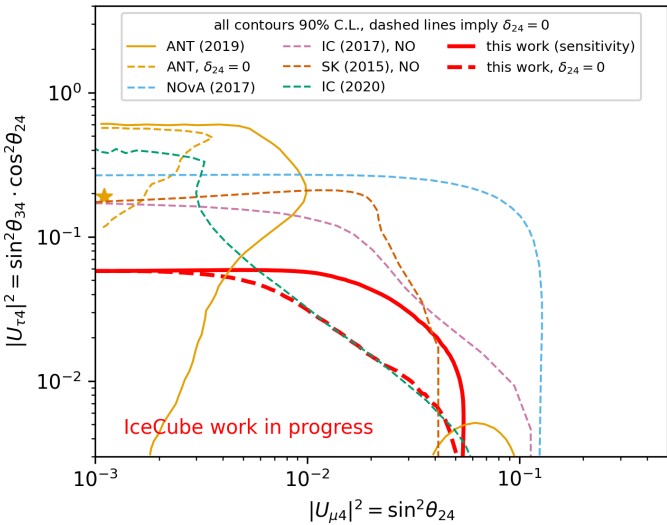

**Figure 2.** Summary of the constraints on light sterile neutrinos from atmospheric and accelerator neutrino experiments. In this figure, the lines are valid for mass-squared-differences greater than 1 eV². The orange star corresponds to the best-fit point obtained by the ANTARES collaboration (ANT). Reproduced from [58].

## 4. Searches with Atmospheric Neutrinos above 100 GeV in Neutrino Telescopes

As previously discussed, at energies above 100 GeV, light sterile neutrinos undergo a resonant conversion due to the presence of matter effects for parameters compatible with the MiniBooNE and LSND observations [43–46,48,50,59]. Two analyses have been performed by the IceCube collaboration to search for this resonant depletion. The first analysis searching for sterile neutrinos using neutrino telescope data was performed by IceCube and published in 2016 [60]. This analysis used approximately 20,000 muon-neutrino events to search for muon-neutrino disappearance due to a light sterile neutrino. The second such analysis was recently unblinded and used approximately 300,000 muon-neutrino events taken over eight years [55,56].

The main difference between these two analyses is the following. First, the event selection of the new analysis was made more efficient than the previous while maintaining a similar level of purity. Second, the treatment of systematic uncertainties was improved compared to the first analysis; all systematic treatments are discussed in [55]. The most impactful systematic change was an improved treatment of the atmospheric neutrino uncertainties.

Previously, this was incorporated by performing the analysis using different, discreet atmospheric neutrino models. The more recent analysis allows continuous changes in the atmospheric neutrino flux. The uncertainties on the atmospheric neutrino flux have two origins: the hadronic interactions models that dictate the yield of the different mesons in the atmospheric shower and the uncertainty in the primary cosmic-ray spectra. The results of the last analysis are shown in Figure 3.

The most recent analysis found a preferred region at the 90% C.L. for large mass-squared-differences. Although this region is not significant, it is compatible with the expectations from global fits to light sterile neutrinos [34,35,61,62]. It is one of the largest muon-neutrino disappearance results compatible with them that has been observed to date.

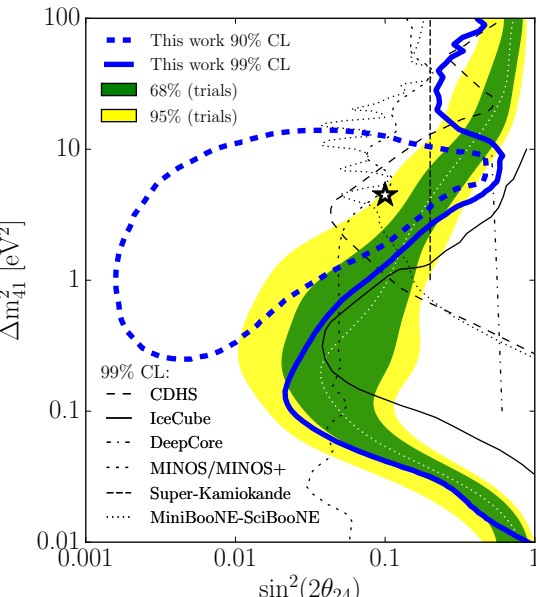

**Figure 3.** In blue solid (dashed): bound at 99% (90%) C.L. for the IceCube results of the search for sterile neutrinos [56]. To illustrate the results by CDHS [10,63], IceCube [60], DeepCore [53], Minos/Minos+ [64], Super-Kamiokande [7], and MiniBooNE-SciBooNE [3,9,65] are shown. The star marks the analysis best-fit point location. Figure from [56].

Beyond the main result shown in Figure 4, the IceCube collaboration performed an analysis looking for heavy sterile neutrinos motivated by hints seen in the one-year data set [54]. In the case of heavy sterile neutrinos—as discussed in the previous section—the primary observable is a unique shape in the angular distribution of the events. This analysis found no significant distortion due to a light sterile neutrino, and constraints were placed in the muon and tau-neutrino mixings, which were parameterized by a set of rotations, see [34] for the relationship between the angles and the mixing parameters.

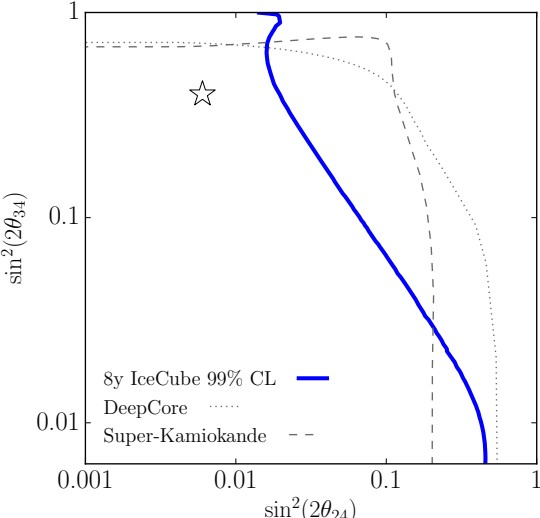

**Figure 4.** In blue solid: 99% C.L. exclusion in the regime of fast oscillations, i.e., heavy sterile neutrinos. The star shows the best fit, and results from other experiments are also shown [7,53].

## 5. Signals in the Astrophysical Neutrino Flavor

The relative amount of the number of electron, muon, and tau event types, also known as the flavor content of the astrophysical neutrinos, was proven to be extremely sensitive to test new physics [66,67]. The flavor of neutrinos in neutrino telescopes can be

distinguished by the morphology of the light deposited in the detector. Muons produced in muon-neutrino charged-current interactions produce long Cherenkov light depositions called tracks.

Neutral-current neutrino interactions, charged-current electron-neutrino interactions, and most of the interactions of tau-neutrinos produce a cascade of particles with an approximately spherical light emission called cascades. Tau-neutrino charged-current neutrinos can be singled-out if the tau produced is boosted enough so that its production and decay point can be resolved; this morphology is known as a double cascade morphology and was recently observed by IceCube [68]. The IceCube collaboration firmly established the flux of high-energy astrophysical neutrinos by measuring all of these morphological channels: cascades [69], starting-events [70], and northern hemisphere tracks [71].

To predict the expected flavor composition of astrophysical neutrinos, one needs to input two things: the flavor composition at the sources and the neutrino oscillation parameters [72]. Since the production mechanism of astrophysical neutrinos is unknown, we do not have sufficient information to assess if neutrinos would maintain their quantum coherence. Additionally, we do not know the spatial distribution of the sources, which makes the distance traversed by the neutrino unknown.

However, we know that the high-energy astrophysical neutrino flux is predominantly from extra-galactic origin [73] from studies looking for clustering around the galactic plane. Given the present energy resolution of astrophysical neutrinos and the expected ratio of baseline to energy, neutrino oscillations will be averaged out. In this regime, the expected flavor composition of astrophysical neutrinos in the presence of a light sterile neutrino can be computed using the following equation [74]

$$P_{\alpha\beta} = \sum_{i=1}^{4} |\mathbf{U}_{\alpha i}|^2 |\mathbf{U}_{\beta i}|^2, \tag{1}$$

where $\mathbf{U}$ is the Pontecorvo–Maki–Nakagawa–Sakata (PMNS) matrix [75,76] extended to include four flavors. In the case where the sterile neutrino is heavier than the parent particle, presumably a pion or kaon on standard production scenarios, the summation on Equation (1) only runs over the light active flavors. A schematic view of how to represent the neutrino flavor composition can be seen in Figure 5, right.

The astrophysical flavor ratio has been inferred using this information, and current measurements are shown in Ref. [68]. Unfortunately, given the current sample size, most of the flavor compositions are still allowed; only extreme scenarios, e.g., a single flavor-dominated scenario, are ruled out. The expected progress in measuring the astrophysical neutrino flavor composition was reported in Ref. [72]. In the next twenty years, with the inclusion of water-, ice-, and mountain-based neutrino detectors—such as KM3NeT [77], GVD [78], P-ONE [79], TAMBO [80], and IceCube-Gen2 [81]—the astrophysical neutrino flavor ratio will be measured with enough precision that the different neutrino production mechanisms will be able to be disentangled [72].

The effect of light sterile neutrinos in the astrophysical neutrino flavor composition was first discussed in Ref. [82], where the authors considered two scenarios. On the first scenario, they assumed that astrophysical neutrinos are produced by conventional means, e.g., from pion decay, and that the sterile neutrino effects are produced only through oscillations. In this scenario, when restricting to mixings compatible with the MiniBooNE and LSND anomalies [34,35,61,62], the effect is small. The second scenario considered a dark matter to be composed of a heavy neutrino that then decayed into a mostly sterile neutrino component.

This second scenario has a more dramatic impact on the expected flavor ratio at Earth; in particular, due to the weak constraints on the heavy neutrino mixing with tau flavors, a large amount of tau flavor composition at Earth is possible. Ref. [74] studied the effect of light sterile neutrinos in an LSND-MiniBooNE-agnostic scenario. In this analysis, the light sterile neutrino mixing parameters were allowed to vary within constraints on the unitarity of the PMNS matrix reported in [83]. The result of this analysis is shown in Figure 5, right.

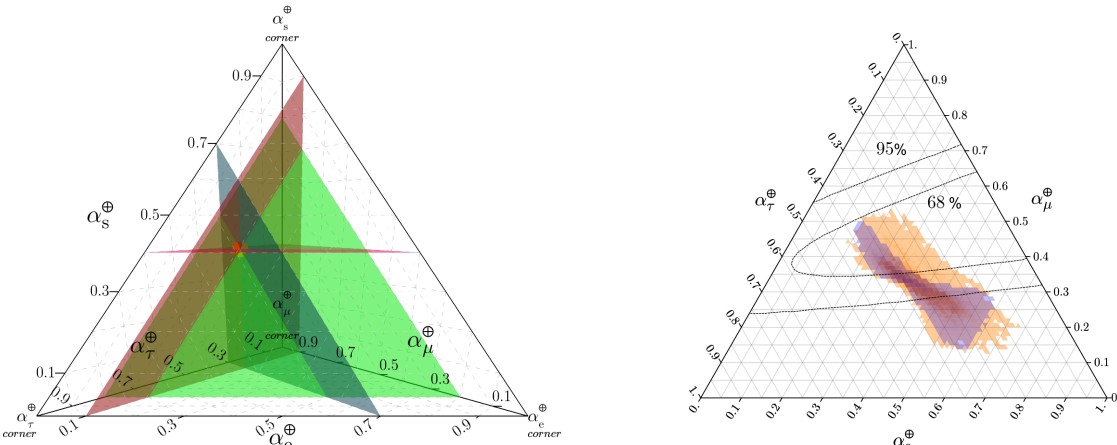

**Figure 5. Left**: a schematic view of how a new sterile state will contribute in to the flavor composition. The distance of every point inside the tetrahedron to the vertexes or the equivalently to the opposite faces will add to one and, therefore, be a good representation for the flavor ratio with one extra sterile state. **Right**: Effect projected in the active states flavor triangle for the case imposing unitarity and without the constraint (for illustration, the contours measured by IceCube are also included) [84]. Figure from [74].

Finally, measuring the flavor content has important implications; however, this will require higher statistical samples of astrophysical neutrino events and perhaps a better event classification. Next generation neutrino telescopes—such as KM3NeT [77], GVD [78], P-ONE [79], TAMBO [80], and IceCube-Gen2 [81]—are going to be essential in answering these questions.

## 6. Conclusions and Future Perspectives

Currently, the LSND and later MiniBoNE neutrino anomalies remain an open problem. Resolving this mystery will require experiments from different fronts, as each provides complementary information beyond the experiments discussed here, including ongoing and planned reactor measurements and accelerator-based experiments, such as the Fermilab short-baseline program [85]. In this scenario, neutrino telescopes provide a unique way to study light sterile neutrinos due to their sheer size. These gigaton-scale detectors can make precision measurements of the atmospheric neutrino spectra at the tens of GeV energy range while observing the high-energy part of the spectra.

The latter can access a unique signature in the search for light sterile neutrinos, namely the matter-induced-resonance disappearance, which, if observed, would provide a smoking gun for light sterile neutrinos. Finally, neutrino telescopes provide a new window to study the Universe with high-energy astrophysical neutrinos. The flavor composition of these neutrinos brings information on the neutrino mixing parameters making them sensitive to additional neutrino mass states. Given all of the above and the advent of new observatories, we expect that neutrino telescopes will continue playing a key role in resolving the light sterile neutrino puzzle.

**Funding:** This research received no external funding.

**Data Availability Statement:** The data presented can be found in the corresponding references.

**Acknowledgments:** We thank to Jeff Lazar for careful reading of this manuscript. CAA is supported by the Faculty of Arts and Sciences of Harvard University, and the Alfred P. Sloan Foundation. JS is supported by the European ITN project H2020-MSCAITN-2019/860881-HIDDeN, the Spanish grants FPA2016-76005-C2-1-P, PID2019-108122GBC32, PID2019-105614GB-C21.

**Conflicts of Interest:** The authors declare no conflict of interest.

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
