# Peer review of "Sterile Neutrinos with Neutrino Telescopes"

_universe, doi:10.3390/universe7110426_

Round 1

Author Response

We would like to thank the referee for the useful and constructive comments on the draft.

We address the following points and modify accordingly the text:

The sterile neutrinos both from the experimental and theoretical point of view is a “hot” subject which has led to numerous citations over past decades. I recommend the authors to enrich their references including also some more general articles related to neutrino physics [e.g. Phys. Rep. 928, 1 (2021)]. It would perhaps better define the context of the present work and help more casual readers. 

We agree a general review article may help to set the context of the topic, specially with a short paper, we add a couple of references including the proposed by the referee. 

Some minor points that should be also corrected are:

  • Page 1, line 2: Although the abbreviations LSND and MiniBooNe correspond to two widespread experimental campaigns, the full names should be specified the first time that the aforementioned abbreviations appear in the text. 

Corrected

  • Page 2, line 51, “…the a light sterile neutrino gives… -> … a light sterile neutrino gives … 

Corrected

  • Page 2, line 59: “…whole energy spectrum [28–30]” -> A full stop is needed after the squared brackets. 

Corrected

  • Figure 2: A star is present in the Figure, but it is not stated in the text and/or the caption at which kind of analysis this marker corresponds to. It should be specified that the marker corresponds to the ANTARES result. 

We have added in the caption that the star corresponds to the ANTARES best-fit point.

  • Page 4, 4 lines from the top of the page: “The results of the latter analysis is shown …” -> “The results of the latter analysis are shown…” 

corrected

  • Page 4, 3 lines from the bottom of the page: “…shown in fig.??,…” -> The Figure number is missing. I assume that is Fig.4?

Corrected

  • Page 5, line 40: “…Additionally, we also do not know…” -> “…Additionally, we do not know…” 

corrected

  • Page 5, line 49: “…the PMNS matrix extended…” -> Please define the abbreviation PMNS and provide the appropriate references. 

We now define it in the text.

  • Page 5, line 62: “…neutrino production mechanics…”->“…neutrino production mechanisms…”

Corrected

Reviewer 2 Report

This paper draft gives a succinct review of the search for sterile neutrinos using neutrino detectors, focusing mainly on the search results and sensitivities with the IceCube detector. I think it is suitable for publication with a few minor changes needed:

1) When writing about the future perspectives in this field, this paper draft mainly focuses on the gigaton-scale detectors, such as KM3NeT, GVD, P-One, IceCube-Gen2, they are all water/ice Cherenkov detectors. I think it's worthwhile to add a short paragraph commenting on any expected updated results from the reactor-based and accelerator-based neutrino experiments in the future, just to form a more complete picture.

2) Other small issues:
On page 2, line 51, "the a" -> "a"
Page 4, line 14, figure number missing.
Page 4, line 24, mislabeling of Figure in the text, should be Figure 3. 
Page 5, line 25, prefer -> preferred
Page 5, line 65, the assume -> they assume

Author Response

We would like to thank the referee for the careful reading of the paper and the useful comments. 

In the following we addressed all the points brought up by the referee.

1) When writing about the future perspectives in this field, this paper draft mainly focuses on the gigaton-scale detectors, such as KM3NeT, GVD, P-One, IceCube-Gen2, they are all water/ice Cherenkov detectors. I think it's worthwhile to add a short paragraph commenting on any expected updated results from the reactor-based and accelerator-based neutrino experiments in the future, just to form a more complete picture.

We have now included references to other experiments.

2) Other small issues:

On page 2, line 51, "the a" -> "a"

corrected

Page 4, line 14, figure number missing.

Corrected

Page 4, line 24, mislabeling of Figure in the text, should be Figure 3. 

Corrected

Page 5, line 25, prefer -> preferred

Corrected

Page 5, line 65, the assume -> they assume

Corrected

Reviewer 3 Report

The authors provide a clear and concise review of the current status of searches for light sterile neutrinos with neutrino telescopes. The paper is well written and I suggest its publication after implementing the following small changes.

In general, please check the tenses.

Line 34: I would remove the comma after “standard”.

Line 43: in Refs. [17-24] 

Line 49: In the range 10^3 GeV - 10^4 GeV 

Line 49-50: Please rephrase this sentence

Line 51: a the light sterile neutrino

Line 54: the other -> the second

Line 59: missing full-stop

Line 66: Please rephrase the second part of the sentence. For example, instead of “motivated by LSND and MiniBooNE” I would prefer “compatible with the LSND and MiniBooNE data” or something similar.

Line 67: at

Line 69: netrino with mass -> neutrino with a mass
Line 74: add a reference for the CP-violating phases

Line 76: constrains -> bounds

Figure 3: Please, rephrase the caption so that each citation corresponds to the experiment.

Figure4: In the caption replace CL with C.L.

Page 4 Line 14: check the figure reference

Page 5 

line 18: replace "result" with "analysis" 

line 34: was proved to be extremely sensitive to test new physics.

line 53: Ref. [42]

line 62: Ref. [48]

line 64: Ref [54]

line 65: they assume
Page 6

line 68-68: The second scenario considered a dark matter to be composed of a heavy neutrino…

Author Response

We would like to thank the referee for the careful reading and corrections on the text. 

In general, please check the tenses.

We have reviewed the English of the whole manuscript.

Line 34: I would remove the comma after “standard”.

Corrected

Line 43: in Refs. [17-24] 

Corrected

Line 49: In the range 10^3 GeV - 10^4 GeV 

Corrected

Line 49-50: Please rephrase this sentence

Corrected

Line 51: a the light sterile neutrino

Corrected

Line 54: the other -> the second

Corrected

Line 59: missing full-stop

Corrected

Line 66: Please rephrase the second part of the sentence. For example, instead of “motivated by LSND and MiniBooNE” I would prefer “compatible with the LSND and MiniBooNE data” or something similar.

Corrected

Line 69: netrino with mass -> neutrino with a mass

Corrected 

Line 74: add a reference for the CP-violating phases

Corrected 

Line 76: constrains -> bounds

Corrected

Figure 3: Please, rephrase the caption so that each citation corresponds to the experiment.

Corrected

Figure4: In the caption replace CL with C.L.

Corrected

Page 4 Line 14: check the figure reference

Corrected

line 18: replace "result" with "analysis" 

Corrected

line 34: was proved to be extremely sensitive to test new physics.

Corrected

line 53: Ref. [42]

Corrected

line 62: Ref. [48]

Corrected

line 64: Ref [54]

Corrected

line 65: they assume

Page 6

Corrected

line 68-68: The second scenario considered a dark matter to be composed of a heavy neutrino…

Corrected

Reviewer 4 Report

Sterile neutrinos are theoretically well – motivated particles but, they remain invisible from the recent large – scale experimental facilities. However, observed experimental anomalies provide hints of their existence and their participation in neutrinos oscillations. Nowadays, several underground, underwater and under-ice facilities are trying to shed some light on these neutrinos that remain neutral under weak interaction.

The authors in this short-review article present an overview of the light sterile neutrinos searches in different neutrino telescopes worldwide. The importance of these facilities in measurements related to the flavor of neutrinos is also highlighted.

To my opinion, the article is scientifically sound, well – written, interesting even for a reader non-specialist in neutrino physics and appropriate for Universe. The current state of the knowledge of the topic including the recent progress and perspectives is concisely described. Taking into consideration all the above, I consider the present review article worth publishing in Universe.

A few minor suggestions to be considered by the authors can be found below:

  • As a reader I would like to see also a deeper theoretical discussion however, I understand that this may be beyond the scope of the present article. Therefore, I would include a few more lines about the sterile neutrino models and if they can -or not- adequately describe all the observed anomalies.
  • I suggest to define Δm2ij and sin22θij providing also short comments about their physical meaning. A schematic description would be also helpful. The importance of squared mass difference in neutrinos searches may be highlighted. Please take into account that Universe readers are distributed in a wide range of scientific topics.
  • A second affiliation is present but nobody seems to be affiliated with this.
  • Page 2, third paragraph, line 51: …the a light sterile…→…a light sterile…
  • Page 4, 3 lines from the end (line 14): figure number is missing
  • Page 5, 2nd paragraph of chapter 5, line 49: …PMNS matrix… → … Pontecorvo–Maki–Nakagawa–Sakata (PMNS) matrix ...
  • Page 5, 2nd paragraph of chapter 5, line 52: …An schematic… → … A schematic ...

Author Response

We would like to thanks the referee for the positive comments and careful reading of the text, 

In the following we address the points brought up by the referee.

As a reader I would like to see also a deeper theoretical discussion however, I understand that this may be beyond the scope of the present article. Therefore, I would include a few more lines about the sterile neutrino models and if they can -or not- adequately describe all the observed anomalies.

We add extra text to shortly comment the theoretical models

I suggest to define Δm2ij and sin22θij providing also short comments about their physical meaning. A schematic description would be also helpful. The importance of squared mass difference in neutrinos searches may be highlighted. Please take into account that Universe readers are distributed in a wide range of scientific topics.

We modified the text to make it more pedagogical.

A second affiliation is present but nobody seems to be affiliated with this.

Corrected

Page 2, third paragraph, line 51: …the a light sterile…→…a light sterile…

Corrected

Page 4, 3 lines from the end (line 14): figure number is missing

Corrected

Page 5, 2nd paragraph of chapter 5, line 49: …PMNS matrix… → … Pontecorvo–Maki–Nakagawa–Sakata (PMNS) matrix …

Corrected

Page 5, 2nd paragraph of chapter 5, line 52: …An schematic… → … A schematic …

Corrected